# An assessment model of bio-efficiency for container terminals in the presence of air emissions

**Long Van Hoang[1], Lan Thi Tuyet Ngo[2,3]*, Ha Thi Pham[3]**

1 Faculty of Management, Ho Chi Minh City University of Law, Ho Chi Minh, Vietnam, 2 Applied Economics Research Group, Dong Nai Technology University, Bien Hoa, Vietnam, 3 Faculty of Economics-Management, Dong Nai Technology University, Bien Hoa, Vietnam

* ngothituyetlan@dntu.edu.vn

## Abstract

Atmospheric pollutants from container terminal operations have adverse effects on the environment in port regions, leading to increased health risks, including respiratory and cardiovascular diseases among local residents. This paper aims to assess bio-efficiency for container terminals (CTs) in the presence of air emissions utilizing a slacks-based measure (SBM) model. In doing so, the paper first adopts cluster analysis to elect homogeneous CTs that aligns with the assumption of DEA theory, then uses a new method to estimate air emissions generated by CTs' operations at harbor zones. Next, the SBM model is used to estimate the bio-efficiency of CTs in the presence of air emissions. Finally, CTs in the Ba Ria-Vung Tau port authorities (PAs) are employed as an empirical study to verify the proposed research model. The proposed research framework can contribute a methodological reference towards the relevant literature on abating atmospheric pollutants in maritime regions.

## 1. Introduction

Container terminals (CTs) are specialized facilities located inside ports to transfer cargo containers between different modes of transportation, such as ships, trucks, and trains nationwide or worldwide. These CTs also serve as exceedingly critical logistic hubs in the global supply chain networks for the efficient movement of goods and the provision of value-added services, such as customs clearance [1], consolidation and deconsolidation of goods [2], inspection and repair of containers [3]. Accordingly, Park, Lee [4] argued that efficient CTs might contribute significantly to the economy and be key drivers of regional and national economic growth.

It has been postulated that CTs discharge a huge amount of air emissions during their operations, primarily resulting from fossil fuel combustion in ships, trucks, terminal equipment, and other machinery. Nonetheless, methods to calculate air emissions to minimize environmental impacts at CTs are still debatable. For instance, Merico, Donateo [5] adopted the DOAS remote sensing technique to measure and monitor air emissions from various sources at CTs (i.e., vessels, and cargo-handling devices) in a Mediterranean harbor in Italy. In the meantime, Zhang, Gu [6] adopted the Ship Traffic Emission Assessment (STEA) approach to

**Data availability statement:** All relevant data are within the manuscript and its Supporting Information files.

**Funding:** The author(s) received no specific funding for this work.

capture air emissions at the Nanjing Longtan Container Port. It can be said that the primary disadvantages of such methods are that they are cost-intensive and time-consuming, as well as require very detailed information on every stage of CTs' lifecycle. To tackle these challenges, this current research introduces a new method to measure air emissions discharged by cargo-handling equipment at CTs.

Currently, two primary approaches have been used to estimate efficiency scores of decision-making units (DMUs) in the presence of air emissions: The radial and non-radial DEA models. The former treats air emissions as a regular input in the production function of DMUs, while the latter considers them a bad output. However, scholars heavily criticize the radial DEA model since it assumes that the efficiency score is measured by proportionally reducing (or increasing) all inputs (or outputs) simultaneously, while maintaining the same proportions among them. In contrast, the non-radial DEA model (i.e., SBM) calculates DMUs' operating efficiency with the assumption of non-proportionate changes of inputs and outputs (hereafter *factors*). Accordingly, this approach is arguably suited for production systems, for instance, CTs.

To fill the literature gaps, the present article aims at assessing bio-efficiency for CTs within a specific port in the presence of air emissions. In doing so, the paper first introduces an innovative technique to figure out the amount of air emissions discharged by CTs. Then, the SBM model in the presence of air emissions is constructed to determine efficiency measures for CTs. Ultimately, CTs affiliated with the Ba Ria-Vung Tau port authorities (hereafter the BR-VT case) are empirically surveyed to verify the suggested research framework. It is worth noting that according to the Ministry of Transport [7], CTs in the Ba Ria-Vung Tau port had been planned to become a regional logistics hub in Vietnam's southern area by 2030. As a result, empirical results are supposed to supply theoretical and practical information for port governments in developing container seaports all over Vietnam.

## 2. Literature review

### 2.1. Relevant literature

As port operations generate a considerable amount of air emissions and cause tremendous effects on the environment surrounding port areas, researchers have paid more attention to integrating them into DEA models to measure the efficiency scores of ports.

The first study using the DEA model to evaluate ports' efficiency was conducted by Roll and Hayuth [8]. It is argued that environmental factors should be adopted to figure out efficiency of ports for better policies. Yet, this study's weakness is to use fictitious data sources to estimate the relative efficiency scores of ports. Tongzon [9] measured the relationship between factors and cargo terminals' efficiency using a two-stage least squares (TSLQ). Similar to Roll and Hayuth [8], Tongzon [9] also suggest using biological variables in ascertaining CTs' performance and efficiency.

When evaluating performance measures of container seaports in East Asia, Chin and Low [10] found that seaports' performance can be changed dramatically when biological factors are utilized in the assessment model. It is also argued that the inclusion of externality mitigation strategies might substantially influence the performance measures of seaports in the sample. Lee, Yeo [11] attempted to evaluate the environmental efficiency of thirteen port cities using the SBM model considering air emissions, such as NOx, SO2, and CO2. It is shown that Singapore, Busan, Rotterdam, Kaohsiung, Antwerp, and New York are highly environmentally efficient, while Hong Kong, Tianjin, Hamburg, Los Angeles, and Jeddah are relatively less environmentally efficient. Besides, this paper calculated air emissions that should be reduced to make port cities become fully efficient. Additionally, this paper presented social costs and opportunity costs for dealing with air emissions in inefficient port cities.

Na, Choi [12] adopted an inseparable input–output SBM model to estimate the environmental efficiencies of eight container ports in China from 2005 to 2014. Empirical results show that the operating performance of Chinese container ports is relatively low, with a mean efficiency score of around 0.6 and there exists minor differences in ports' efficiency among different regions in China. This finding is somewhat consistent with Chin and Low [10]. Tsao and Thanh [13] developed a multi-objective mixed robust possibilistic flexible programming approach to assess sustainable seaport-dry port network (SSDPN) design under an uncertain environment. Empirical results determine the optimal number, location, and capacity of dry ports in designing SSDPN for minimizing the economic costs and environmental and social impacts. In addition, not only do numerical analyses reduce the total network costs by up to 1.14%, but also improve efficiency of the average computational time for large-sized networks in the context of uncertainty.

Wang, Zhou [14] conducted a green efficiency evaluation to improve Chinese ports' operating performance by the integration of a cross-efficiency model and the Tobit regression, with nitrogen oxides (NOX) and sulfur oxides (SOX) being used as undesirable outputs of port activities. Results are in agreement with Na, Choi [12], when arguing that Chinese ports' green efficiency is relatively low and port developments are unbalanced among investigated regions during the five-year period. Furthermore, the Tobit regression analysis finds a significant relationship between economic development, industrial structure, and ports' green efficiency scores.

According to recent research, operational efficiency criteria, for instance, turnaround time [15], cargo throughput [16], and berth productivity [17] are increasingly being used to assess terminal performance. Besides, sustainability indicators have been incorporated to assess environmental concerns about waste management [18], energy use [19], and air emissions [20]. Notably, contemporary research has shown that maximizing terminal throughput and lowering greenhouse gas emissions present two challenges, for example, the higher volume of cargo in a shorter amount of time [20] and the adoption of sustainable solutions [19]. Accordingly, developing holistic models that simultaneously optimize efficiency and minimize environmental footprints is highly demanding.

Cui, Chen [21] conducted an evaluation and analysis of the green efficiency of China's coastal ports under the "double carbon" goal by two improved DEA models with CO2 emissions. It is illustrated that external environmental variables influence the efficiency measurement of ports' green development significantly; thus, excluding these effects helps increase the green efficiency values of most ports in China. Further, similar to what had been done by Na, Choi [12] and Wang, Zhou [14], the overall green efficiency of ports in China is comparatively low and tended to go downward from 2012 to 2020. Hsu, Huynh [22] assessed the operating performance and efficiency of container seaports in the port of Kaohsiung (Taiwan) using the DEA model considering air pollutants as a undesirable output. It is argued that CTs' managers should reduce air emissions by at least 15%–20% to boost CTs' efficiency measures, and align with sustainable development goals for the whole port.

## 2.2. Research gaps

Although previous studies have assessed port efficiency by the DEA model with the integration of air pollutants generated by CTs, some research gaps should be identified to be filled in the current article.

To begin, when applying the assessment model of efficiency, the immense majority of prior studies flouted a homogeneous characteristic of DMUs, which is one of the basic assumptions of the DEA theory. Golany and Roll [23] argued that the application of the DEA model must satisfy three primary conditions: (1) deploying similar technology, processes, or production

methods; (2) using similar types of inputs and producing similar types of outputs so that the comparison of efficiency across DMUs is meaningful; (3) operating in similar business environments, including regulatory, economic, and market conditions so that external factors do not disproportionately influence the efficiency scores of certain DMUs.

Second, some existing research did not meet the general rule-of-thumb for DEA application, which illustrates that the number of DMUs under evaluation should be at least two-three times the number of factors. Banker, Charnes [24] and Hsu, Huynh [22] explained that DEA is a non-parametric method; thus, the model's discriminatory power increases with the DMUs' expansion. More specifically, in the case of a relatively small ratio of DMUs involved in comparison with factors, many DMUs can be fully efficient simply because the DEA model lacks the resolution to differentiate between them.

At last, the vast majority of relevant literature, when adopting air emissions as an undesirable output in the evaluation model of efficiency, often merely collected data sources from DMUs' self-announcements. In other words, the accuracy of the information provided by DMUs is hard to be verified in practice since DMUs may underreport their air emissions to satisfy environmental regulations set up by governments. Further, the data on the release of air pollutants into the atmosphere from various operations of CTs is usually unavailable, especially in least-developed and developing countries. To fill this gap, the current paper introduces a new method for calculating the amount of air emissions generated by CTs at port areas.

## 3. Research methods

### 3.1. Research process

Fig 1 outlines a structured research process of the paper. After defining the research objectives, this paper constructs the SBM model to estimate the bio-efficiency score for CTs in the presence of air emissions. Once homogeneous DMUs (i.e., CTs) are determined by cluster analysis, the factors of CTs' operations are defined, thanks to experts' consultation and prior research. Then, this research collects data collection for the SBM model from the Ba Ria-Vung Tau port authorities (PAs) and Vietnam Maritime Administration. Afterwards, the study calculates the amount of $CO_2$ emissions emitted by CTs. Before running the SBM model, an isotonicity test is performed to confirm the significant relationship between inputs and outputs. Following this, an overall bio-efficiency analysis and slacks analysis are conducted to assess the performance of the DMUs. Finally, improvement policies for inefficient DMUs are suggested based on empirical results.

### 3.2. The SBM model with air emissions

This section presents the establishment of the SBM model to calculate CTs' bio-efficiency in the presence of air emissions. The SBM model can be formed, as follows:

**Data sources.**

$i(1,2,...,I)$ Index of inputs

D: The number of desirable outputs, for instance, container throughput.

$d(1,2,...,D)$ Index of desirable outputs.

U: The number of bad outputs, for example, air emissions.

$u(1,2,..,U)$ Index of bad outputs.

J: The number of DMUs, viz., CTs.

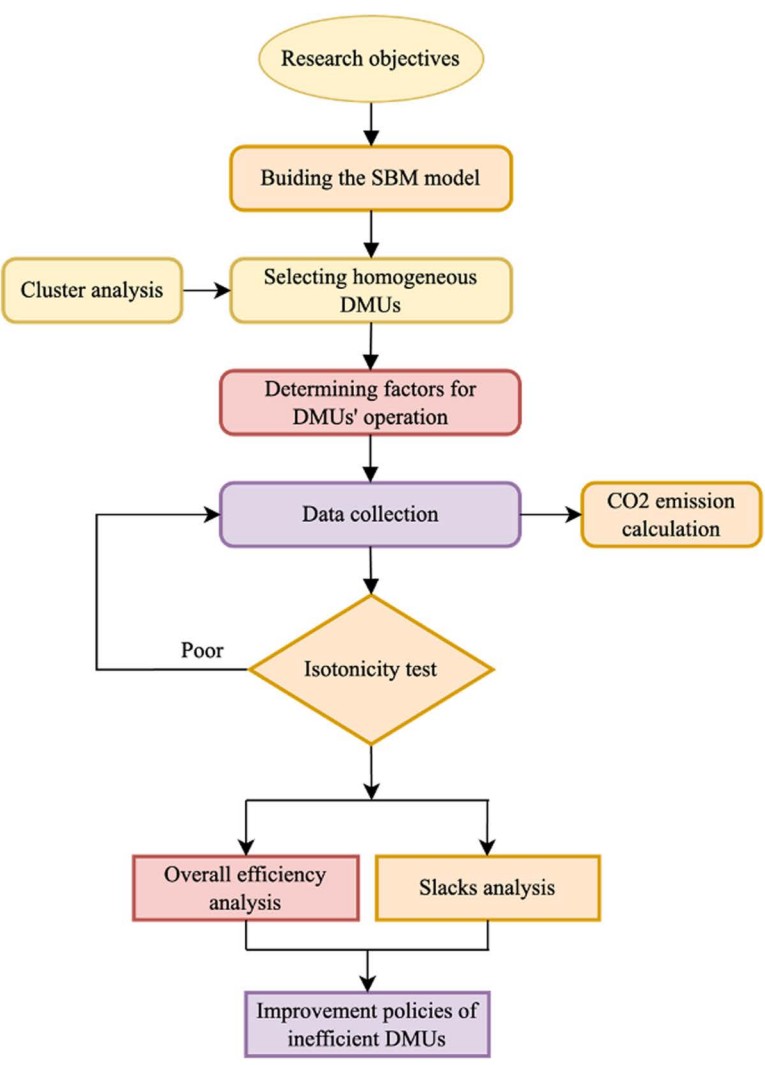

**Fig 1. Process of bio-efficiency analysis for CTs.**

$j(1,2,...,J)$ Index of DMUs.

0: Index of particular container terminal, whose bio-efficiency score is being estimated.

$x_{ij}$: Surveyed amount of input i of DMU j.

$y_{uj}$: Surveyed amount of bad output u of DMU j.

$y_{dj}$: Surveyed amount of desirable output d of DMU j.

**Variables.**

$(\lambda_1, \lambda_2, ..., \lambda_J)$: Non-negative parameters adopted for computing a linear aggregate of the CTs in the data sample.

$S_{d0}^+$: Slack variables of desirable output $d$ of DMU $0$.

$S_{u0}^{-}$: Slack variables of bad output $u$ of DMU $0$.

$S_{i0}^{-}$: Slack variables of input $i$ of DMU $0$.

The proposed SBM model to calculate CTs' bio-efficiency in the presence of air emissions can be formed, as below [25]:

$$
h_0^* = \min \frac{1 - \dfrac{1}{I}\displaystyle\sum_{i=1}^{I}\dfrac{S_{i0}^{-}}{x_{i0}}}{1 + \dfrac{1}{D+U}\left(\displaystyle\sum_{d=1}^{D}\dfrac{S_{d0}^{g+}}{y_{d0}^{g}} + \displaystyle\sum_{u=1}^{U}\dfrac{S_{u0}^{b-}}{y_{u0}^{b}}\right)}
$$

$$
x_{i0} = \sum_{j=1}^{J} x_{ij}\lambda_j + S_{i0}^{-}
$$

$$
y_{d0}^{g} = \sum_{j=1}^{J} y_{dj}^{g}\lambda_j - S_{d0}^{g+} \tag{1}
$$

$$
y_{u0}^{b} = \sum_{j=1}^{J} y_{uj}^{b}\lambda_j + S_{u0}^{b-}
$$

$$
S_i^{-} \geq 0,\ S_d^{g+} \geq 0,\ S_u^{b-} \geq 0,\ \lambda_j \geq 0
$$

$$
i = 1,2,...,I;\ j = 1,2,...,J;\ d = 1,2,...,D;\ u = 1,2,...,U.
$$

It would be worth noting that the efficiency score for $DMU_0$, as attained by Model (1), is computed on the condition of constant returns to scale. Thence, such a score is referred to the overall efficiency (OE), which integrating technical efficiency and scale efficiency.

Suppose that the optimal solution of Model (1) is $\left(\lambda^{*}, S_i^{-*}, S_u^{-*}, S_d^{+*}\right)$. Then, we have:

Theorem 1: *CTs are defined bio-efficient if and only if $h^* = 1$. In this case, the value of slacks for inputs, desirable outputs and air emissions equals zero.*

If $h^* < 1$, CTs will not be bio-efficient. In this case CTs can use the following SBM-projection to become bio-efficient:

$$
\begin{cases}
\check{x}_{i0} = x_{i0} - s_{i0}^{-*} \\
\check{y}_{u0} = y_{u0} - s_{u0}^{-*} \\
\check{y}_{d0} = y_{d0} - s_{d0}^{+*}
\end{cases} \tag{2}
$$

## 3.3. The selection of DMUs for the SBM model

According to Hsu, Huang [16], cluster analysis is necessary for DEA to ensure that DMUs being compared are homogeneous, which leads to more accurate and meaningful efficiency assessments. By grouping similar DMUs together, cluster analysis helps eliminate the noise arising from comparing heterogeneous DMUs. As noted earlier, the present article uses the cluster analysis approach to elect homogeneous CTs for the efficiency evaluation model. In this article, eighteen CTs in the BR-VT case are chosen for the empirical case.

The process to select homogeneous CTs is explained:

**Step 1**: Determining variables for clustering.

The goal of cluster analysis is to group CTs into clusters based on their similarities, so that CTs within each cluster are homogeneous regarding the selected variables. Thanks to that, this analysis will identify CTs operating under similar conditions, allowing for more meaningful comparisons and benchmarking. In doing so, thanks to CTs' operating features [26, 27]

and expert consultation, the current paper selects fourteen variables for the cluster analysis, including: Management practices, technological innovation, customer satisfaction, environmental practices, operational flexibility, technological innovation, safety culture, corporate social responsibility, strategic alliances and partnerships, reputation, customs and regulatory relations, service differentiation, crisis management, cultural alignment, and marketing strategies.

**Step 2**: Obtaining experts' rating:

Suppose that $a_{ij}(i=1,2,...,I)$ is the $i^{th}$ criterion of the $DMU_j(j=1,2,...,J)$. The value of $a_{ij}$ is rated by industrial experts working at CTs using the five-point Likert questionnaire. Let $k(k=1,2,...,K)$ is the group of experts in the survey. And each expert rating creates a single matrix $A^k = \left[a_{ij}\right]_{I \times J}, k=1,2,...,K$. Using the arithmetic mean $a_{ij} = \sum_{k=1}^{K} a_{ij}^k / K$, we can form the integrated matrix $A = \left[a_{ij}\right]_{I \times J}$.

**Step 3**: Conducting the cluster analysis:

*First of all,* this paper determines the optimal number of clusters by the gap statistic using some algorithms [28, 29]:

(1) Let $k = (1,2,...,k_{max})$ the number of clusters. Then, the pooled within-cluster sum of squares around the cluster means is computed by $W_k = \sum_{r=1}^{k} \left(\frac{1}{2n_r}\right) \cdot \Sigma_{ij} \left(x_{ij} - x_{ij'}\right)^2$.

(2) Creating N cluster from $k = (1,2,...,k_{max})$ Such N clusters' gap statistics are estimated by
$$Gap_n(k) = E_n^* \log(W_k) - \log(W_k)$$

(3) Let $\overline{w} = \left(\frac{1}{B}\right) \Sigma_b \log(W_{kb}^*)$. Calculate the standard deviation
$$sd(k) = \sqrt{\left(\frac{1}{b}\right) \Sigma_b \left(\log(W_{kb}^*) - \overline{w}\right)}, \text{ and then } s_k = sd_k \times \sqrt{1 + \frac{1}{B}}$$

(4) The optimal number of clusters will satisfy the condition of $Gap(k) \geq Gap(k+1) - s_{k+1}$

By such a process, the paper determined two optimal clusters from 18 CTs, as seen in Fig 2.

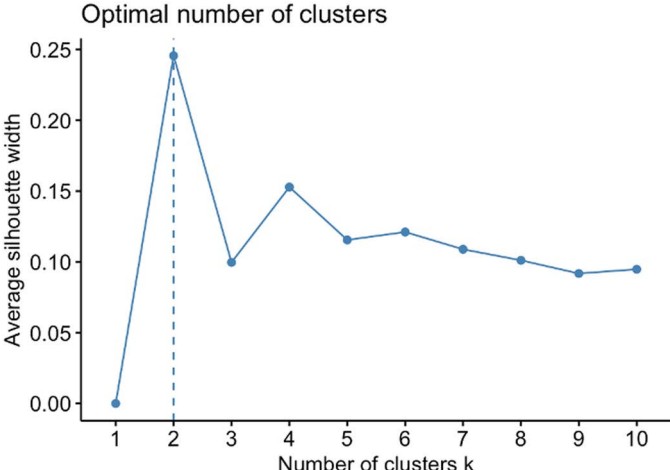

**Fig 2. Optimal numbers of clusters.**

Secondly, the paper assigns data observations to clusters employing a hierarchical agglomerative algorithm once two optimal clusters are identified by gap statistics, as mentioned above. The main goal of this step is to allocate data observations to pre-determined clusters. This step starts with $N$ data points. Then, we combine the two most similar clusters to result in $N-1$ clusters. This process involves an interactive approach, and will end when $N$ data points are assigned to only one cluster.

Applying the aforementioned processes, the cluster analysis results are illustrated in Fig. 3. Evidently, 18 CTs are categorized into two groups. More particularly, Group A includes twelve CTs while Group B comprises five CTs. In principle, CTs located in the same group might be considered homogeneous [26, 27]. To conclude, the current article chooses 12 CTs in Group A as comparable DMUs to verify the proposed research model, as exhibited in the Fig 3.

### 3.4. The calculation of CO2 emissions

The total amount of CO2 emissions discharged by CTs are estimated on the basis of air emissions emitted by different pieces of equipment, such as ship-to-shore cranes, trucks, straddle carriers, reach stackers, etc. Call h and w be the number of pieces of equipment at CTs and types of modalities, respectively. As such, CO2 emissions discharged by *xth* CT (named $M_x$) are figured out by:

$$M_x = \sum_{i=1}^{h}\sum_{j=1}^{w}\left(v_{ij} * f_D + P_{ij} * f_E\right) \qquad (3)$$

In which:

$$v_{ij} = n_{ij}\left(C_{ij} + c_{ij} * \bar{X}_{ij}\right); \forall ij \in T \qquad (4)$$

$$P_{ij} = n_{ij} * p_{ij}; \forall ij \in T \qquad (5)$$

Where: $v_{ij}$ and $P_{ij}$ are the liter of diesel and kWh of electricity used by the *ith* equipment for the *jth* modality, respectively. Meanwhile $f_D = (= 2.26)$ and $f_E (0.832)$ are the emission factors of diesel and electricity, respectively.

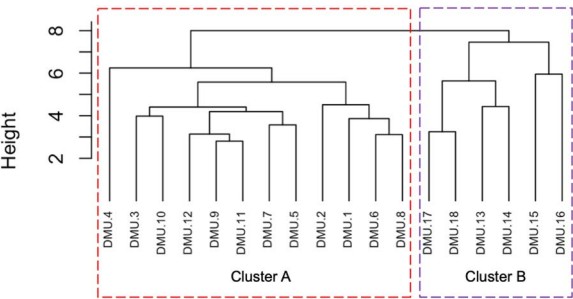

**Fig 3. Cluster classification for the BR-VT case.**

## 4. Empirical case

### 4.1. Input and output items for CTs' operation

Golany and Roll [23] posited that the accuracy and reliability of DEA are highly sensitive to the choice of factors for DMUs' operations. Besides, Kao [30] argued that the DEA model can result in efficiency scores that do not represent the true performance of the DMUs in some cases, whose factors do not accurately reflect the real operational processes of DMUs. Thanks to the relevant literature, six factors are employed to estimate bio-efficiency for CTs for the BR-VT case, as exhibited in Table 1.

### 4.2. Data collection

After identifying all factors for CTs' operations, the research team collected data for the SBM model from the Ba Ria-Vung Tau PAs and cross-checked gathered information with Vietnam Maritime Administration's annual port trade records if necessary. As a result, we have the data source, as presented in Table 2. For the BR-VT case, applying Formula (4) ~ (6), the amount of $CO_2$ emissions emitted by CTs is also estimated.

**Table 1. Input and output factors for CTs' operation.**

| Factors | | Unit of measurement | Explanation | Reference |
|---|---|---|---|---|
| Inputs | Employees | Number of employees | Total number of personnel employed at the terminal. | Dimitriou [38], Davarzani, Fahimnia [39] |
| | STS cranes | Number of cranes | Total number of Ship-to-Shore (STS) cranes available at the terminal. | Ding, Jo [40], Geerlings and Van Duin [41] |
| | Yard cranes | Number of cranes | Total number of yard cranes (including RTGs and RMGs) available for container handling. | Li, Seo [42], Wiegmans and Witte [43] |
| | Container yard | m2 | The total area of the container yard, measured in square meters. | Park, Mohamed Abdul Ghani [44], Tsao and Thanh [13] |
| Out-puts | Air emissions | kg/TEU | Amount of air pollutants emitted per TEU (Twenty-Foot Equivalent Unit) handled. | Na, Choi [12], Roy, De Koster [26] |
| | Container throughput | TEUs | Total number of TEUs handled by the terminal in a given period | Pérez, González [45], Wanke, Nwaogbe [46] |

**Table 2. The data source for the BR-VT case.**

| DMUs | Employees | Container yard (m2) | STS cranes | Yard cranes | Air emissions (kg/TEU) | Container throughput (TEUs) |
|---|---|---|---|---|---|---|
| CT.1 | 208 | 362,753 | 17 | 44 | 97.8 | 1,124,122 |
| CT.2 | 115 | 136,942 | 10 | 16 | 54.7 | 266,583 |
| CT.3 | 78 | 151,131 | 8 | 21 | 45.7 | 714,838 |
| CT.4 | 143 | 230,261 | 10 | 19 | 56.3 | 513,295 |
| CT.5 | 238 | 902,329 | 22 | 31 | 76.8 | 1,180,129 |
| CT.6 | 175 | 351,182 | 9 | 39 | 59.4 | 196,132 |
| CT.7 | 83 | 92,972 | 7 | 13 | 40.2 | 112,135 |
| CT.8 | 64 | 232,133 | 13 | 24 | 51.2 | 128,910 |
| CT.9 | 47 | 149,045 | 10 | 11 | 83.8 | 152,441 |
| CT.10 | 207 | 451,358 | 23 | 36 | 93.3 | 1,090,609 |
| CT.11 | 89 | 127,310 | 12 | 21 | 56.2 | 134,102 |
| CT.12 | 153 | 331,280 | 11 | 25 | 80.4 | 410,913 |

### 4.3. Isotonicity test

The application of DEA requires that the relationship between inputs and outputs is not erratic. In other words, increasing the value of any input, while keeping other factors constant, should not decrease any output, but should instead lead to an increase in the value of at least one output. This feature is also a basic consumption of the DEA theory [31, 32]. To test isotonicity, this paper statistically assessed the relationship between inputs and outputs by calculating the Pearson correlation coefficients between each input and output pair. As demonstrated in Table 3, a strong positive correlation between inputs and outputs suggests that isotonicity is being maintained.

### 4.4. Overall bio-efficiency

The overall bio-efficiency for the BR-VT case is displayed in Table 4. It is evident that the bio-efficiency of the BR-VT case is rather low, with an average score of approximately 0.405, meaning that CTs are not operating at their optimal capacity. Additionally, only two out of 12 CTs (equivalent to 16.7%) achieve overall bio-efficiency, comprising CTs 3 and 5, which might act as the peer group (or optimal *targets*) for bio-inefficient CTs [23, 25]. The peer group measures how bio-inefficient CTs can improve their bio-efficiency scores, by reducing their inputs

**Table 3. Isotonicity test.**

|  | Employees | Container yard (m2) | STS cranes | Yard cranes | Air emissions (kg/TEU) | Container throughput (TEUs) |
|---|---|---|---|---|---|---|
| Employees | 1 | 0.822** | 0.729** | 0.798** | 0.595* | 0.786** |
| Container yard (m2) |  | 1 | 0.791** | 0.586* | 0.502* | 0.724** |
| STS cranes |  |  | 1 | 0.606* | 0.695** | 0.791** |
| Yard cranes |  |  |  | 1 | 0.558* | 0.634* |
| Air emissions (kg/TEU) |  |  |  |  | 1 | 0.620** |
| Container throughput (TEUs) |  |  |  |  |  | 1 |

Note:

* and

**: Significant level of 0.05 and 0.01, respectively

**Table 4. Overall bio-efficiency and slacks for the BR-VT case.**

| DMUs | Overall bio-efficiency | Peer group | Slacks | | | | | |
|---|---|---|---|---|---|---|---|---|
|  |  |  | Employees | Container yard (m2) | STS cranes | Yard cranes | Air emissions (kg/TEU) | Container throughput (TEUs) |
| CT.1 | 0.604 | CT.3 (1.57) | 85 | 125,091 | 4 | 11 | 26 | 0 |
| CT.2 | 0.270 | CT.3 (0.37) | 86 | 80,581 | 7 | 8 | 38 | 0 |
| CT.3 | 1.000 | CT.3 (1) | 0 | 0 | 0 | 0 | 0 | 0 |
| CT.4 | 0.462 | CT.3 (0.72) | 87 | 121,740 | 4 | 4 | 23 | 0 |
| CT.5 | 1.000 | CT.5 (1) | 0 | 0 | 0 | 0 | 0 | 0 |
| CT.6 | 0.113 | CT.3 (0.27) | 154 | 309,716 | 7 | 33 | 47 | 0 |
| CT.7 | 0.148 | CT.3 (0.16) | 71 | 69,264 | 6 | 10 | 33 | 0 |
| CT.8 | 0.107 | CT.3 (0.18) | 50 | 204,879 | 12 | 20 | 43 | 0 |
| CT.9 | 0.199 | CT.3 (0.21) | 30 | 116,816 | 8 | 7 | 74 | 0 |
| CT.10 | 0.556 | CT.3 (1.53) | 88 | 220,782 | 11 | 4 | 24 | 0 |
| CT.11 | 0.123 | CT.3 (0.19) | 74 | 98,958 | 10 | 17 | 48 | 0 |
| CT.12 | 0.272 | CT.3 (0.57) | 108 | 244,405 | 6 | 13 | 54 | 0 |

and air emissions or increasing their outputs, to become overall bio-efficient. Moreover, Hsu and Huynh [33] also illustrated that inefficient CTs with $h^* < 1$ are not performing at an optimal level, implying that these CTs can reduce inputs and emissions without cutting of outputs. For example, the bio-efficiency value of CT.1 is 0.604, demonstrating that this DMU wasted 39.6% (100% - 60.4%) of its essential input factors to produce final products, indicating room for improvement.

## 4.5. The slack variable analysis

The slacks of factors for the BR-VT case are presented in Table 4. In theory, slacks refer to the excess amounts of inputs and air emissions or the shortfall quantities of outputs preventing CTs from being overall bio-efficiency [1, 27]. Based on such slacks, deploying Formula (2), bio-inefficient CTs can become overall bio-efficient. Table 5 also shows optimal values (projection) of factors for CT.1. The calculation of projection for remaining CTs can be performed in a similar way.

Slack results also argue that air emissions are one of the key factors of inefficiencies that should be treated in CTs. Liu, Guo [34] demonstrated that dealing with air emissions places a financial burden on firms (i.e., CTs), which can be defined as opportunity costs (OCs) involving the trade-offs associated with implementing measures to reduce air emissions versus the potential benefits that might be gained if those resources are allocated elsewhere. Liu, Guo [34] suggested an equation to estimate such opportunity costs: $OCs = (1 - h^*) \times revenue$. Revert to CT.1 as a typical example, its total revenues are reported as $1,623,031 in 2023. Accordingly, its OCs are estimated as $642,720. The remaining CTs' OCs can be figured out in the same vein.

## 4.6. Discussions

This paper assesses bio-efficiency for container terminals in the presence of air emissions by using the SBM model, with the BR-VT case as an empirical case. Not only do results present bio-efficiency scores of CTs, but also provide the way for CTs to become overall bio-efficient. In practice, findings can supply managerial information for port stakeholders (i.e., terminal investors) in appraising and selecting efficient terminal projects.

The bio-efficiency scores, as presented in Table 4 indicate the relatively low efficiency of CTs for the BR-VT case. Among 12 evaluated CTs, just two CTs (i.e., CTs 3 and 5) are overall bio-efficient, with a score of 1.000, implying they are operating on the efficiency frontier. In contrast, other CTs exhibit varying degrees of bio-inefficiency, with CT.6 (0.113), CT.8 (0.107), and CT.7 (0.148) being the least bio-efficient. Stated differently, these low scores suggest that such CTs have substantial room for improvement in optimizing their resource use and reducing air emissions.

Under our investigation, bio-efficient CTs (i.e., CTs 3 and 5) have been implementing energy efficiency programs, such as transition to renewable energy sources, to low air

**Table 5. Projection for CT.1.**

|  | Slacks | Original values | Optimal | % change |
|---|---|---|---|---|
| Employees | 85 | 208 | 123 | −41.0 |
| Container yard (m2) | 125091 | 362,753 | 237,662 | −34.5 |
| STS cranes | 4 | 17 | 13 | −26.0 |
| Yard cranes | 11 | 44 | 33 | −24.9 |
| Air emissions (kg/TEU) | 26 | 98 | 72 | −26.5 |
| Container throughput (TEUs) | 0 | 1,124,122 | 1,124,122 | 0 |

emissions. Evidently, CT. 3 has air emissions of 45.7 kg/TEU, much lower than CT.1 (97.8 kg/TEU) and CT.10 (93.3 kg/TEU). In addition, some low-carbon technologies, for instance, solar panels, electric yard tractors, and LED lighting, have been deployed by these DMUs since 2022.

## 4.7. Sensitivity analysis

It will be worth noting that bio-efficiency scores of CTs are estimated from a relatively small sample size, thus might introduce variability and bias, potentially leading to less reliable efficiency scores. Accordingly, this study checks the sensitivity of empirical findings by adopting bootstrap resampling, which repeatedly samples from the original dataset with replacement to create numerous new datasets, each of the same size as the original [35]. The step to carry out bootstrap resampling is as follows:

Step 1 is to randomly sample with replacement from the original dataset to create $N$ new datasets. Note that the larger $N$ is, the better the sensitivity results are [36].

Step 2 is to calculate bio-efficiency scores for each resampled dataset using Model (1).

Step 3 is to analyze the distribution of these efficiency scores computed in Step (2) to understand variability and confidence intervals. Then, we determine the bias of efficiency

by $Bias\left(\rho_j\right) = E\left(\rho_j\right) - \rho_j = B^{-1}\sum_{b=1}^{B}\left(\hat{\rho}_{jb}\right) - \rho_j$ and estimate the bias-corrected efficiency by

$\tilde{\rho}_j = \rho_j - Bias\left(\rho_j\right) = 2\rho_j - B^{-1}\sum_{b=1}^{B}\left(\hat{\rho}_{jb}\right)$. Lastly, the $\left(1-\alpha\right)\%$ confidence interval of $\rho_j$ is esti-

mated by $2\rho_j - \hat{\rho}_{(1-\alpha)} \leq \rho_j \leq 2\rho_j - \hat{\rho}_{(\alpha)}$.

Table 6 illustrates sensitivity analysis for the BR-VT case. Evidently, the bias of the bi-efficiency scores is relatively low, ranging from −0.028 to 0.007. Thanks to that, the discrepancy in bio-efficiency between the original dataset and resampled ones is inconsiderable. To sum up, performing the sensitivity analysis by bootstrap resampling assesses the robustness of the bio-efficiency scores and verifies that empirical findings are still reliable regardless of the small sample size.

**Table 6. Sensitivity analysis for the BR-VT case.**

| DMUs | DMUs | Bias | Bias-corrected efficiency | Standard Deviation | 2.50% | 97.50% |
|------|------|------|---------------------------|--------------------|-------|--------|
| | | | | | Bias-corrected confidence interval | |
| CT.1 | 0.604 | 0.002 | 0.602 | 0.039 | 0.550 | 0.664 |
| CT.2 | 0.27 | 0.002 | 0.268 | 0.037 | 0.243 | 0.297 |
| CT.3 | 1 | −0.028 | 1.000 | 0.035 | 1 | 1.000 |
| CT.4 | 0.462 | −0.001 | 0.462 | 0.027 | 0 | 0.504 |
| CT.5 | 1 | −0.027 | 1.000 | 0.034 | 0.910 | 1.000 |
| CT.6 | 0.113 | 0.003 | 0.110 | 0.029 | 0.103 | 0.124 |
| CT.7 | 0.148 | 0.001 | 0.147 | 0.009 | 0.135 | 0.163 |
| CT.8 | 0.107 | 0.007 | 0.100 | 0.050 | 0 | 0.118 |
| CT.9 | 0.199 | −0.001 | 0.199 | 0.012 | 0.179 | 0.217 |
| CT.10 | 0.556 | −0.001 | 0.556 | 0.033 | 1 | 0.606 |
| CT.11 | 0.123 | 0.005 | 0.118 | 0.039 | 0.112 | 0.135 |
| CT.12 | 0.272 | −0.002 | 0.272 | 0.017 | 0.245 | 0.297 |

# 5. Conclusions

## 5.1. Conclusions

It has been demonstrated that overall bio-efficient CTs often consume less energy and produce fewer emissions per container handled, thus aligning with global trends toward sustainable development of the port industry. Nonetheless, calculating the amount of air emissions in portp areas and selecting the policies to improve the overall bio-efficiency of CTs are still debatable. To fill the literature gaps, the current article aims at assessing bio-efficiency for CTs within a specific port in the presence of air emissions using the SBM model. Some theoretical and practical contributions to the relevant literature are as follows:

Theoretically, cluster analysis assists in electing homogeneous CTs to meet the basic assumption of the DEA theory. It has been argued that homogeneity is a crucial characteristic of DMUs, implying that all evaluated DMUs carry out similar functions [24], operate under similar conditions [37], and use comparable inputs to produce comparable outputs [23]. To put it another way, DMUs should be similar regarding their operational environment and the types of operations they perform. Previous research admitted the non-homogeneous DMUs cannot only lead to biased benchmarking [33], but also invalid comparison [30]. To cope with such a barrier, the current article conducted cluster analysis to satisfy comparable CTs for the SBM model. It can be said that this approach will be applicable to the selection of homogeneous DMUs in many different areas, such as banking, education, and engineering.

Adopting the SBM model to estimate the bio-efficiency of CTs in the presence of air emissions enables rigorously assessment of CTs' performance. It has been posited that CTs' operations not only create desirable outputs (i.e., container throughputs), but also discharge a vast amount of air emissions from fuel combustion that can adversely affect air quality and contribute to environmental issues, for instance, smog, acid rain, and climate change. Therefore, integrating air emissions into the evaluation model of bio-efficiency allows to rank CTs' performance fairly for resource allocation. It is believed that the SBM model can be implementable for the evaluation of operating performance and capacities of transport systems, such as road networks, air transportation, and public transit.

It is widely accepted that air emissions are often unavailable, especially in the least developed and developing nations, as environmental regulations are less stringent or poorly enforced. To contribute to the relevant literature, this paper adopted formulas to calculate air pollutants discharged by CTs' operations via the amount of energy consumed and the traveling distance of cargo-handling equipment. It can be argued that the methods in this article can be feasible in quantifying atmospheric pollutants for manufacturing systems, such as factories, inland container depots, and distribution centers.

According to practical perspectives, the proposed research model provides CTs with a framework to evaluate and improve their bio-efficiency score, mainly by reducing air emissions, thus helping CTs minimize environmental impacts at port regions to satisfy global sustainability goals. With the BR-VT case, the findings show two overall bio-efficient CTs, which serve as optimal benchmarks for inefficient ones. Besides, slacks of the SBM model provide practical information for terminal managers in improving bio-efficient CTs. On top of that, the empirical findings might enable Ba Ria-Vung Tau PAs to make strategic actions and prioritize the allocation of investment to bio-efficient CTs.

## 5.2. Research limitations

Some potential research limitations should be addressed, as follows. First of all, although CTs' operations release many types of undesirable outputs in addition to air emissions, the proposed research model merely focuses on air emissions, potentially overlooking other

significant environmental impacts, such as water pollution, waste management, and noise pollution, which should be considered by future studies for a more comprehensive assessment of CTs' bio-efficiency. Secondly, prior literature admitted that CTs' bio-efficiency scores fluctuate over time due to various factors, such as changes in technology [22], operational practices [34], regulations, and market conditions [14, 21]. Accordingly, assessing bio-efficiency for CTs at a single point of time, as done in this paper, disables the capture of these dynamic changes. Hence, this limitation leaves a room for longitudinal studies to explore what specific factors (e.g., technological advancements, policy changes, or economic shifts) contribute to changes in CTs' bio-efficiency over time.

## Supporting information

**S1 Data.** **The file contains data relevant to the paper.**
(DOCX)

## Author contributions

**Conceptualization:** Lan Thi Tuyet Ngo.

**Data curation:** Lan Thi Tuyet Ngo.

**Formal analysis:** Lan Thi Tuyet Ngo.

**Investigation:** Lan Thi Tuyet Ngo.

**Methodology:** Lan Thi Tuyet Ngo.

**Software:** Long Van Hoang.

**Supervision:** Long Van Hoang.

**Validation:** Long Van Hoang.

**Visualization:** Long Van Hoang.

**Writing – original draft:** Long Van Hoang.

**Writing – review & editing:** Long Van Hoang, Ha Thi Pham.

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
