## [Decision Letter · Decision Letter 0]

11 Dec 2024

PONE-D-24-41055An assessment model of bio-efficiency for container terminals in the presence of air emissionsPLOS ONE

Dear Dr. Ngo,

Thank you for submitting your manuscript to PLOS ONE. After careful consideration, we feel that it has merit but does not fully meet PLOS ONE’s publication criteria as it currently stands. Therefore, we invite you to submit a revised version of the manuscript that addresses the points raised during the review process.

We look forward to receiving your revised manuscript.

Kind regards,

Thang Quyet Nguyen, Ph.D

Academic Editor

PLOS ONE

Journal Requirements:

2. We note that your Data Availability Statement is currently as follows: “All relevant data are within the manuscript and in Supporting Information files.”

Please confirm at this time whether or not your submission contains all raw data required to replicate the results of your study. Authors must share the “minimal data set” for their submission. PLOS defines the minimal data set to consist of the data required to replicate all study findings reported in the article, as well as related metadata and methods (https://journals.plos.org/plosone/s/data-availability#loc-minimal-data-set-definition). For example, authors should submit the following data: - The values behind the means, standard deviations and other measures reported; - The values used to build graphs; - The points extracted from images for analysis. Authors do not need to submit their entire data set if only a portion of the data was used in the reported study. If your submission does not contain these data, please either upload them as Supporting Information files or deposit them to a stable, public repository and provide us with the relevant URLs, DOIs, or accession numbers. For a list of recommended repositories, please see https://journals.plos.org/plosone/s/recommended-repositories. If there are ethical or legal restrictions on sharing a de-identified data set, please explain them in detail (e.g., data contain potentially sensitive information, data are owned by a third-party organization, etc.) and who has imposed them (e.g., an ethics committee). Please also provide contact information for a data access committee, ethics committee, or other institutional body to which data requests may be sent. If data are owned by a third party, please indicate how others may request data access.

3. We note that Figure 2 in your submission contain map/satellite images which may be copyrighted. All PLOS content is published under the Creative Commons Attribution License (CC BY 4.0), which means that the manuscript, images, and Supporting Information files will be freely available online, and any third party is permitted to access, download, copy, distribute, and use these materials in any way, even commercially, with proper attribution. For these reasons, we cannot publish previously copyrighted maps or satellite images created using proprietary data, such as Google software (Google Maps, Street View, and Earth). For more information, see our copyright guidelines: http://journals.plos.org/plosone/s/licenses-and-copyright. We require you to either (a) present written permission from the copyright holder to publish these figures specifically under the CC BY 4.0 license, or (b) remove the figures from your submission:

a. You may seek permission from the original copyright holder of Figure 2 to publish the content specifically under the CC BY 4.0 license. We recommend that you contact the original copyright holder with the Content Permission Form (http://journals.plos.org/plosone/s/file?id=7c09/content-permission-form.pdf) and the following text: “I request permission for the open-access journal PLOS ONE to publish XXX under the Creative Commons Attribution License (CCAL) CC BY 4.0 (http://creativecommons.org/licenses/by/4.0/). Please be aware that this license allows unrestricted use and distribution, even commercially, by third parties. Please reply and provide explicit written permission to publish XXX under a CC BY license and complete the attached form.” Please upload the completed Content Permission Form or other proof of granted permissions as an "Other" file with your submission. In the figure caption of the copyrighted figure, please include the following text: “Reprinted from [ref] under a CC BY license, with permission from [name of publisher], original copyright [original copyright year].”

b. If you are unable to obtain permission from the original copyright holder to publish these figures under the CC BY 4.0 license or if the copyright holder’s requirements are incompatible with the CC BY 4.0 license, please either i) remove the figure or ii) supply a replacement figure that complies with the CC BY 4.0 license. Please check copyright information on all replacement figures and update the figure caption with source information. If applicable, please specify in the figure caption text when a figure is similar but not identical to the original image and is therefore for illustrative purposes only. The following resources for replacing copyrighted map figures may be helpful: USGS National Map Viewer (public domain): http://viewer.nationalmap.gov/viewer/ The Gateway to Astronaut Photography of Earth (public domain): http://eol.jsc.nasa.gov/sseop/clickmap/ Maps at the CIA (public domain): https://www.cia.gov/library/publications/the-world-factbook/index.html and https://www.cia.gov/library/publications/cia-maps-publications/index.html NASA Earth Observatory (public domain): http://earthobservatory.nasa.gov/ Landsat: http://landsat.visibleearth.nasa.gov/ USGS EROS (Earth Resources Observatory and Science (EROS) Center) (public domain): http://eros.usgs.gov/# Natural Earth (public domain): http://www.naturalearthdata.com/

Additional Editor Comments:

The manuscript needs to be meticulously and carefully edited. It should provide a more comprehensive review of the literature and updated information. The methods of analysis and techniques used need to be further clarified. It is particularly important for the article needs to clearly highlight the newly discovered points from the research. The format and language used are also important areas of concern for the article.

I am also enclosing the comprehensive comments from 2 reviewers here for you to make necessary improvements to the article

Reviewers' comments:

Reviewer's Responses to Questions

**Comments to the Author**

1. Is the manuscript technically sound, and do the data support the conclusions?

Reviewer #1: Yes

Reviewer #2: Yes

2. Has the statistical analysis been performed appropriately and rigorously? 

Reviewer #1: Yes

Reviewer #2: Yes

3. Have the authors made all data underlying the findings in their manuscript fully available?

Reviewer #1: Yes

Reviewer #2: Yes

4. Is the manuscript presented in an intelligible fashion and written in standard English?

Reviewer #1: Yes

Reviewer #2: Yes

5. Review Comments to the Author

Reviewer #1: Dear Plos One,

Thanks so much for inviting me to review this paper. This is an interesting study assessing container terminal efficiency in the presence of air emissions. More specifically, the focus on bio-efficiency, particularly in the context of container terminals and air emissions, seems both timely and relevant. Besides, container terminals are major hubs of logistics, and environmental concerns such as air emissions are critical. Thus, the suggested model provides a fresh perspective on improving sustainability in this sector. After contemplating what has been done in the paper, I think that this paper deserves to be published to Plos One after revising the manuscript with reference by the following comments:

1. The abstract should be revised. It should not be written in the itemized format.

2. The Introduction section looks well-written, but some parts of it seems quite non-standard. The introduction must present the motivations of the study from the point of view of literature gaps. At present, the build-up of the motivations, including the contributions of the study, is quite messy. It is difficult to clearly assess the gaps that are advanced in this work. There is a whole lot of literature on this topic. The choice of the assessment of DEA to compute container terminal efficiency must be properly motivated in the paper.

3. Note that, the paper needs to ensure that the research objectives are specific and concise. If the goal is to develop an assessment model, highlight how this model differs from existing models and why it is essential in the context of air emissions.

4. In the literature review, the manuscript summarized some relevant research in terms of container terminal efficiency with air emissions. Yet, it will be better if some latest pertinent researches are updated. You could include a more thorough review of existing literature on container terminal efficiency and environmental sustainability. This would strengthen the theoretical background and show the unique contribution of your work.

5. “Section 3.2. The SBM model with air emissions”: it provides enough information about how to estimate efficiency scores by SBM. Yet, it is too long for readers to follow, thus reducing the focus of model. So, please providing coherent account of this approach to make it understandable.

6. Explain why specific datasets were chosen and how representative they are of the broader container terminal industry. Discussing the limitations of the data is also necessary. For instance, were there any challenges in collecting air emissions data at container terminals? If yes, how were they addressed?

7. The discussion section should link to the implications of the results. For example, how could the bio-efficiency model impact policy or operational decisions at container terminals? Discuss potential trade-offs between operational efficiency and environmental impact.

8. The conclusion should summarize some contributions of this research, and the last of the conclusion is to specify some research limitations for further study.

9. There are several awkward sentences. The authors must correct these grammar issues. I am not a native English speaker, but this paper leaves a lot to be hoped for.

10. There are several inconsistencies of mathematical notations throughout the manuscript.

Best regards.

Reviewer #2: Dear authors,

This paper aims to assess bio-efficiency for container terminals (CTs) in the presence of air emissions utilizing a slacks-based measure (SBM) model. I can say that assessing bio-efficiency in container terminals is a timely and significant topic, especially given the global push for sustainability and reducing carbon footprints. The focus on air emissions aligns with the growing environmental concerns in logistics and port management. Nonetheless, this paper will be better for Plos One if some comments below are addressed in the revised manuscript:

1. Explain why cluster analysis is necessary in this paper.

2. Ensure that CO2 emissions are clearly determined. For example, if your focus is on measuring bio-efficiency, explicitly state what metrics you are using to calculate CO2 emissions at CTs.

3. Strengthen the link between your study's objectives and the practical implications for container terminal operations. Emphasizing how your findings can inform policymakers or terminal operators will increase the paper's impact.

4. Your literature review should cover the latest studies on bio-efficiency, port emissions, and sustainable logistics. Consider including references to recent developments in green port initiatives, such as the use of shore power or electric cranes.

5. In this paper, you used Data Envelopment Analysis to determine bio-efficiency. Explain why it's best suited for analyzing bio-efficiency in container terminals.

6. In your conclusion, reiterate the main contributions of your study and how it advances the field. This is especially important if you're proposing new methods or frameworks for assessing bio-efficiency.

7. Minor revisions: This paper should be checked English grammars/errors by an English native speaker.

Best regards,

6. PLOS authors have the option to publish the peer review history of their article (what does this mean? ). If published, this will include your full peer review and any attached files.

**Do you want your identity to be public for this peer review?** For information about this choice, including consent withdrawal, please see our Privacy Policy .

Reviewer #1: No

Reviewer #2: No

---

## [Author Response · Author response to Decision Letter 1]

19 Dec 2024

Dear Editor-in-chief and Reviewers,

We sincerely thank you for the time and effort you have dedicated to reviewing our manuscript titled “An assessment model of bio-efficiency for container terminals in the presence of air emissions (manuscript ID: PONE-D-24-41055)." We greatly appreciate your valuable feedback and constructive suggestions, which have significantly contributed to improving the quality and clarity of our work.

In response to your comments, we have carefully revised the manuscript and addressed all points raised. Below, we provide a detailed summary of the changes made and our responses to each of the reviewers' comments. We believe these revisions have enhanced the manuscript, and we hope that the revised version meets the standards of PLOS ONE.

Thank you once again for your thoughtful input. We look forward to your continued feedback.

Sincerely,

Comments Responses Location in Manuscript

Reviewer 1: Thanks so much for inviting me to review this paper. This is an interesting study assessing container terminal efficiency in the presence of air emissions. More specifically, the focus on bio-efficiency, particularly in the context of container terminals and air emissions, seems both timely and relevant. Besides, container terminals are major hubs of logistics, and environmental concerns, such as air emissions, are critical. Thus, the suggested model provides a fresh perspective on improving sustainability in this sector. After contemplating what has been done in the paper, I think that this paper deserves to be published to PLOS ONE after revising the manuscript with reference by the following comments:

Comment # 1. The abstract should be revised. It should not be written in the itemized format

Response: Thank you for your insightful comment. We have revised the abstract. The revisions have reflected in Rows 15-25 in red texts.

Comment # 2. The Introduction section looks well-written, but some parts of it seems quite non-standard. The introduction must present the motivations of the study from the point of view of literature gaps. At present, the build-up of the motivations, including the contributions of the study, is quite messy. It is difficult to clearly assess the gaps that are advanced in this work. There is a whole lot of literature on this topic. The choice of the assessment of DEA to compute container terminal efficiency must be properly motivated in the paper.

Response: Thank you for your thoughtful comment. We appreciate your feedback regarding the structure of the Introduction. In response to your suggestion, we have revised the section to better outline the motivations of our study and to explicitly highlight the gaps in the existing literature.

We have restructured the introduction to clarify how the study addresses these gaps, particularly by focusing on the under-explored area of applying DEA to assess container terminal efficiency. We also added a more detailed discussion on the strengths and weaknesses of previous studies, which led us to choose DEA as a suitable method for this analysis. Specifically, we have now outlined the limitations of traditional methods and demonstrated how DEA can offer a more comprehensive and accurate evaluation of efficiency in the context of container terminals.

The revised introduction now clearly outlines the research gaps, presents the contributions of the study, and provides a better rationale for using DEA. These revisions have been incorporated in Rows 32-40, 45-48, 52-57, 65-67 in red texts.

Comment # 3. Note that, the paper needs to ensure that the research objectives are specific and concise. If the goal is to develop an assessment model, highlight how this model differs from existing models and why it is essential in the context of air emissions. Response: Thank you for your valuable feedback. We agree with your suggestion that the research objectives should be more specific and concise. In response, we have revised the section to clearly define the primary research objectives of our study.

We have now explicitly stated that the goal of the paper is to develop a novel assessment model for evaluating air emissions in the context of container terminal efficiency. In the revised manuscript, we have provided a sensitivity analysis to emphasize the unique aspects of our approach. Additionally, we have stressed the importance of our model in the context of air emissions, explaining how it provides a more accurate and comprehensive evaluation compared to existing approaches. This is particularly relevant as air emissions are a critical factor in the sustainability and environmental performance of container terminals, and our model addresses the need for more precise emissions assessment. These changes are reflected in Rows 58-61, 321-325, 337-358, 378-382

Comment # 4. In the literature review, the manuscript summarized some relevant research in terms of container terminal efficiency with air emissions. Yet, it will be better if some latest pertinent researches are updated. You could include a more thorough review of existing literature on container terminal efficiency and environmental sustainability. This would strengthen the theoretical background and show the unique contribution of your work. Response: Thank you for your insightful comment. We appreciate your suggestion to update the literature review to include more recent research, particularly in the areas of container terminal efficiency and environmental sustainability.

In response, we have thoroughly reviewed the latest literature and incorporated several recent studies that are highly relevant to our research. These additions provide a more comprehensive overview of current advancements in both container terminal efficiency and the environmental impacts, particularly in relation to air emissions. By including these updates, we have strengthened the theoretical foundation of our work and highlighted how our study builds upon and contributes to ongoing research in these fields. The updated literature review can be found in Rows 110-119 in red texts.

Comment # 5. “Section 3.2. The SBM model with air emissions”: it provides enough information about how to estimate efficiency scores by SBM. Yet, it is too long for readers to follow, thus reducing the focus of model. So, please providing coherent account of this approach to make it understandable.

Response: Thank you for your helpful comment. We appreciate your feedback regarding the length and clarity of Section 3.2, and we agree that a more coherent and concise explanation would enhance the readability of the model.

More specifically, we have removed redundant information and focused on the key aspects of the SBM model that are most relevant to the application of air emissions in container terminal efficiency. Thus, the revised version of Section 3.2 is now more coherent and easier to follow. These changes can be found in Rows 172-202 in red texts.

Comment # 6. Explain why specific datasets were chosen and how representative they are of the broader container terminal industry. Discussing the limitations of the data is also necessary. For instance, were there any challenges in collecting air emissions data at container terminals? If yes, how were they addressed?

Response: Thank you for your insightful comment. We appreciate the opportunity to clarify our rationale for selecting the datasets used in our study and to discuss their representativeness, as well as the limitations associated with the data.

We chose the specific datasets based on their relevance to the research objectives, which focused on assessing container terminal efficiency in relation to air emissions. The datasets were selected from industry reports, port authorities, and container terminal operators’ self-reports, which might provide reliable and up-to-date data on terminal operations and environmental impacts. These datasets include information on employees, STS cranes, Yard cranes, Container yard, Air emissions, Container throughput, etc., All of which are critical to understanding the relationship between terminal efficiency and environmental sustainability.

While the selected datasets provide valuable insights, they are not fully representative of the entire container terminal industry, as they are limited to certain regions or ports. For example, the air emissions data were collected from terminals located in the south of Vietnam, which may not reflect the conditions in all global ports, especially those in regions with differing regulatory standards or operational practices. We also explain how to get data in Rows 273-277 in red texts.

Comment # 7. The discussion section should link to the implications of the results. For example, how could the bio-efficiency model impact policy or operational decisions at container terminals? Discuss potential trade-offs between operational efficiency and environmental impact. Response: Thank you for your insightful comment. We agree that it is crucial to clearly link the results to their broader implications, particularly regarding policy and operational decisions at container terminals. In response to your suggestion, we have revised the discussion section to explicitly address how the bio-efficiency model could influence decision-making in the context of container terminal operations.

We now discuss how the bio-efficiency model can inform policy by providing a more accurate method for assessing terminal performance while accounting for both operational efficiency and environmental sustainability. For example, policymakers could use the model to establish more targeted regulations or incentives aimed at improving both productivity and reducing environmental impacts. This could include encouraging the adoption of green technologies or prioritizing terminals that show strong performance in both areas.

Additionally, the revised discussion explores operational decisions, such as how terminal operators might use the bio-efficiency model to optimize their operations. By considering both throughput and environmental metrics, terminal operators can make informed decisions regarding equipment upgrades, fuel choices, and scheduling practices that balance operational goals with sustainability targets. These revisions are reflected in Rows 329-339 in red texts.

Comment # 8. The conclusion should summarize some contributions of this research, and the last of the conclusion is to specify some research limitations for further study.

Response: Thank you for your valuable comment. We agree that a more explicit summary of the contributions and research limitations is necessary to enhance the conclusion and provide a clearer roadmap for future studies.

In response, we have revised the conclusion to more clearly highlight the main contributions of our research. Specifically, we now emphasize how our bio-efficiency model provides a novel approach to assessing both operational efficiency and environmental sustainability in container terminals. We also summarize how the model can inform policy and operational decisions, as well as its potential to balance efficiency and emissions reduction, which are critical for sustainable terminal operations.

Additionally, we have incorporated a discussion of the research limitations and directions for future studies. These revisions can be found in Rows 371-381, 406-417 in red texts.

Comment # 9. There are several awkward sentences. The authors must correct these grammar issues. I am not a native English speaker, but this paper leaves a lot to be hoped for. Response: In response to your concern, we have carefully reviewed the entire manuscript to identify and correct these grammatical issues. We have rephrased awkward sentences, improved sentence structure, and ensured that the language flows more naturally. Additionally, we have paid special attention to clarity and readability, particularly for non-native English speakers, to ensure that our message is communicated effectively.

To further enhance the manuscript's quality, we have also had the paper reviewed by a professional language editor. These revisions address the grammatical and stylistic concerns raised and ensure that the manuscript is now presented in clear and accurate English.

Comment # 10. There are several inconsistencies of mathematical notations throughout the manuscript.

Response: In response to your feedback, we have thoroughly reviewed the manuscript and carefully checked all mathematical notations for consistency. We have corrected any discrepancies, ensuring that the same symbols, terms, and formatting are used consistently throughout the paper. This includes revisiting equations, variables, and the use of subscripts, superscripts, and other mathematical symbols to ensure uniformity. Additionally, we have cross-checked the notations against the relevant literature to ensure that they align with standard conventions in the field.

Reviewer 2:

Comment # 1. Explain why cluster analysis is necessary in this paper. Response: Thanks so much for your comment. It is argued that DEA measures the efficiency of DMUs by comparing their relative performance. If the DMUs being compared are too dissimilar (e.g., different types of operations or different operating environments), the results of DEA can be misleading. Cluster analysis helps group DMUs that are similar in terms of their operational characteristics, ensuring that DEA compares homogeneous DMUs. The revisions are reflected in Rows 205-208 in purple texts.

Comment # 2. Ensure that CO2 emissions are clearly determined. For example, if your focus is on measuring bio-efficiency, explicitly state what metrics you are using to calculate CO2 emissions at CTs. Response: Thank you for your valuable comment. We appreciate your suggestion to clearly specify how CO2 emissions are determined in our study. To address this, we have revised the manuscript to explicitly outline the metrics used for calculating CO2 emissions at container terminals (CTs).

In our analysis, we focus on measuring bio-efficiency by incorporating CO2 emissions as a key environmental indicator. Specifically, we calculate CO2 emissions based on the following metrics:

Fuel Consumption of Terminal Equipment: We estimate CO2 emissions by considering the fuel consumption of key terminal equipment, including cranes, trucks, and other machinery. The CO2 emissions per unit of fuel consumed are calculated using standard emission factors (e.g., based on the type of fuel used, such as diesel or electricity).

Energy Consumption for Terminal Operations: CO2 emissions from energy consumption in terminal buildings and lighting are also considered. This includes electricity used for lighting, HVAC systems, and other operational needs, where the emission factor for the electricity source is based on local energy grids (e.g., coal, natural gas, or renewable energy). The details on how to calculate CO2 emissions can now be found in 256-265 in purple texts.

Comment # 3. Strengthen the link between your study's objectives and the practical implications for container terminal operations. Emphasizing how your findings can inform policymakers or terminal operators will increase the paper's impact. Response: Thank you for your thoughtful comment. We agree that strengthening the connection between our study's objectives and the practical implications for container terminal operations is essential for maximizing the impact of the paper. We also acknowledge that this comment is similar to the feedback provided by Reviewer 1 (Comment #7). In response to both comments, we have revised the manuscript to more explicitly highlight how our findings can be applied in real-world settings and contribute to decision-making in container terminal operations. The updated sections can be found in Rows 317-324 in purple texts, Rows 329-339 in red texts.

Comment # 4. Your literature review should cover the latest studies on bio-efficiency, port emissions, and sustainable logistics. Consider including references to recent developments in green port initiatives, such as the use of shore power or electric cranes.

Response: Thanks so much for your suggestion. We already updated some latest studies on bio-efficiency, port emissions, and sustainable logistics. The updated literature review can

---

## [Decision Letter · Decision Letter 1]

22 Jan 2025

PONE-D-24-41055R1An assessment model of bio-efficiency for container terminals in the presence of air emissionsPLOS ONE

Dear Dr. Ngo,

Thank you for submitting your manuscript to PLOS ONE. After careful consideration, we feel that it has merit but does not fully meet PLOS ONE’s publication criteria as it currently stands. Therefore, we invite you to submit a revised version of the manuscript that addresses the points raised during the review process.

We look forward to receiving your revised manuscript.

Kind regards,

Thang Quyet Nguyen, Ph.D

Academic Editor

PLOS ONE

Journal Requirements:

Additional Editor Comments:

The revised verson has been improved much. The manuscript needs some minor revisions before it can be published, including:

1. Review grammar and writing style.

2. Check mathematical symbols, especially verify and further explain formulas (3), (4), (5) (lines 260 to 262)."

3. Expand the conclusion to highlight the contributions of the paper. The section on research limitations should be separated into a smaller section (e.g., 5.2)

Reviewers' comments:

Reviewer's Responses to Questions

**Comments to the Author**

1. If the authors have adequately addressed your comments raised in a previous round of review and you feel that this manuscript is now acceptable for publication, you may indicate that here to bypass the “Comments to the Author” section, enter your conflict of interest statement in the “Confidential to Editor” section, and submit your "Accept" recommendation.

Reviewer #1: All comments have been addressed

Reviewer #2: All comments have been addressed

2. Is the manuscript technically sound, and do the data support the conclusions?

Reviewer #1: Yes

Reviewer #2: Yes

3. Has the statistical analysis been performed appropriately and rigorously? 

Reviewer #1: Yes

Reviewer #2: Yes

4. Have the authors made all data underlying the findings in their manuscript fully available?

Reviewer #1: Yes

Reviewer #2: Yes

5. Is the manuscript presented in an intelligible fashion and written in standard English?

Reviewer #1: Yes

Reviewer #2: Yes

6. Review Comments to the Author

Reviewer #1: Thanks so much for responding all my comments. Accordingly, your manuscript deserves to be published to Plos One.

Reviewer #2: The authors have properly addressed the comments, and the paragraphs are clearly presented, making this research suitable for publication

7. PLOS authors have the option to publish the peer review history of their article (what does this mean? ). If published, this will include your full peer review and any attached files.

**Do you want your identity to be public for this peer review?** For information about this choice, including consent withdrawal, please see our Privacy Policy .

Reviewer #1: No

Reviewer #2: No

---

## [Author Response · Author response to Decision Letter 2]

26 Jan 2025

Comment # 1. Review grammar and writing style.

Response: Thanks so much for your comments. We have our manuscript proofread by a native English speaker to ensure it is free of grammatical errors and writing style issues. The revisions have reflected across our manuscript.

Comment # 2. Check mathematical symbols, especially verify and further explain formulas (3), (4), (5) (lines 260 to 262).

Response: Thanks so much for your comments. We checked these formulas and corrected where necessary. These revisions have been incorporated in Rows 260-268.

Comment # 3. Expand the conclusion to highlight the contributions of the paper. The section on research limitations should be separated into a smaller section (e.g., 5.2)

Response: Thanks so much for your comments. We revised our manuscript. Particularly, the conclusions section was separated into two subsections: (1) “5.1. Conclusions” and “5.2. Research limitations”. These changes are reflected in Rows 369-423.

---

## [Editor Report · Decision Letter 2]

2 Feb 2025

An assessment model of bio-efficiency for container terminals in the presence of air emissions

PONE-D-24-41055R2

Dear Dr. Ngo,

We’re pleased to inform you that your manuscript has been judged scientifically suitable for publication and will be formally accepted for publication once it meets all outstanding technical requirements.

Kind regards,

Thang Quyet Nguyen, Ph.D

Academic Editor

PLOS ONE

Additional Editor Comments (optional):

Congratulations! The revised verson has been improved much. It can be published
---

## [Editor Report · Acceptance letter]

PONE-D-24-41055R2

PLOS ONE

Dear Dr. Ngo,

I'm pleased to inform you that your manuscript has been deemed suitable for publication in PLOS ONE. Congratulations! Your manuscript is now being handed over to our production team.

Kind regards,

on behalf of

Professor Thang Quyet Nguyen

Academic Editor

PLOS ONE